# Enhancing Immediate Memory, Potential Learning, and Working Memory with Transcranial Direct Current Stimulation in Healthy Older Adults

**DOI:** 10.3390/ijerph191912716

**Published:** 2022-10-05

**Authors:** Encarnación Satorres, Juan C. Meléndez, Alfonso Pitarque, Elena Real, Mireia Abella, Joaquin Escudero

**Affiliations:** 1Department of Developmental Psychology, Faculty of Psychology, University of Valencia, Av. Blasco Ibañez 21, 46010 Valencia, Spain; 2Department of Methodology, Faculty of Psychology, University of Valencia, Av. Blasco Ibañez 21, 46010 Valencia, Spain; 3Hospital General of Valencia, Av. Tres Cruces, 2, 46014 Valencia, Spain

**Keywords:** tDCS, healthy older adults, immediate memory, working memory, learning potential

## Abstract

Background: Transcranial direct current stimulation (tDCS) has emerged as a prevention method or minimizer of the normal cognitive deterioration that occurs during the aging process. tDCS can be used to enhance cognitive functions such as immediate memory, learning, or working memory in healthy subjects. The objective of this study was to analyze the effect of two 20-min sessions of anodal transcranial direct stimulation on immediate memory, learning potential, and working memory in healthy older adults. Methods: A randomized, single-blind, repeated-measures, sham-controlled design was used. The sample is made up of 31 healthy older adults, of whom 16 were in the stimulation group and 15 were in the sham group. The anode was placed on position F7, coinciding with the left dorsolateral prefrontal cortex region, and the cathode was placed on Fp2, the right supraorbital area (rSO). Results: When comparing the results of the treatment group and the sham group, differences were observed in working memory and learning potential; however, no differences in immediate memory were found. Conclusion: The results showed that tDCS is a non-invasive and safe tool to enhance cognitive processes in healthy older adults interested in maintaining some cognitive function.

## 1. Introduction

Cognitive skills play an important role in the daily functioning of older adults. However, some of these cognitive abilities decline during the aging process. In normal aging, one of the main signs of deterioration is memory loss, which affects both immediate and long-term memory [1]. In aging, there is also a decrease in executive functions (EF), that is, higher-level cognitive skills such as planning, problem-solving, and working memory (WM), all of which are necessary to perform daily activities and maintain independence with age [2]. In fact, working memory is related to other frontally controlled cognitive functions such as language or learning, and so stimulating these cognitive functions will produce more complex actions and thoughts [3].

Recently, transcranial direct current stimulation (tDCS) is emerging as a promising technique for cognitive enhancement. tDCS is a non-invasive brain stimulation technique that is increasingly being used to modulate neuronal activity [4]. The action mechanism of tDCS is based on the principles of neuronal plasticity, given that it is induced through the generation of a sub-threshold stimulation polarity-dependent alteration of membrane potentials, modifying spontaneous discharge rates. Cortical excitability is modulated, resulting in either hypopolarization/excitation or hyperpolarization/inhibition, depending on the polarity of the stimulation [5]; cathodal tDCS resulting in a decrease and anodal stimulation in an increase of cortical excitability [6]. It has been shown that during the encoding phase anodic stimulation improved memory performance in a subsequent recognition task. However, cathodal stimulation impaired subsequent recognition of stimuli [7]. Systematic reviews indicate that in the case of anodal tDCS, administration of higher current doses (density and density loading) results in higher percentages of accuracy on cognitive tasks in healthy participants, although these effects are modest [8]. Stimulation can be applied online (during the task) or offline (immediately after the task). Existing findings in the literature do not allow establishing the optimal timing of tDCS administration to induce effects on memory [9]. However, a systematic study suggests that completing the cognitive task during tDCS (online), compared to following tDCS (offline), is to lead to increased accuracy percentages on cognitive tasks [8].

A number of studies have demonstrated that tDCS can be used to enhance cognitive functions in healthy subjects, and it has been employed to improve memory, learning, or working memory. These studies have mainly applied stimulation to the left dorsolateral prefrontal cortex (DLPFC) and they have shown its effectiveness in both healthy subjects and neuropsychiatric samples such as patients with mild cognitive impairment or Alzheimer’s disease [10]. The systematic review and meta-analysis by [11] suggest that tDCS should provide more satisfactory results in populations with pathologies because there may be a ceiling effect in healthy participants.

Some studies confirm the efficacy of anodic tDCS on memory, noting that these effects are due to improved recovery [12] or even consolidation [13]. Other authors point out that better recall performance might be explained by more efficient coding [14]. Application of anodal tDCS over the left DLPFC during the encoding phase can enhance performance on verbal memorization [7]. Other studies in healthy older adults [15] showed that tDCS improved verbal episodic memory, possibly through reconsolidation, similar results to those obtained in healthy young adults [16]. Additionally, it has been providing evidence for the conclusion that the stabilization of episodic memories may be facilitated by the direct interaction of tDCS with the mechanisms of consolidation [17]. A recent meta-analysis reported positive effects of anodic tDCS over DLPFC on episodic memory performances in healthy older adults, showing significant and modest improvements immediately after stimulation and maintenance of the effect after a long period [18]. However, a review has pointed out that there is no effect on immediate memory, and this would support the hypothesis that tDCS mechanisms would facilitate plasticity during learning, and perhaps during consolidation, generating stronger and more persistent memories [19].

Using anodal tDCS can facilitate learning as well as word generation [20]. Additionally, it has been verified that focal anodal tDCS stimulation of the DLPFC increased the rate of verbal learning compared to sham [21] and improvement on learning and memory processes in healthy adults [19]. tDCS has been used in healthy older adults to modulate mainly cognitive and behavioral processes by applying the principles of neuroplasticity and polarity-dependent cortical modifications. However, it is important to identify the area that justifies the application of anodal stimulation to facilitate improvements based on the learning potential. Achieving effects in this process could be decisive in cognitive rehabilitation, especially in neurocognitive disorders such as mild cognitive impairment and early Alzheimer’s disease. It has been suggested that an increase in working memory efficiency may therefore account for the significantly faster rate of learning observed on declarative verbal memory tasks [21]. Results for working memory performance are mixed. In healthy participants, repeated sessions of anodal tDCS over the DLPFC [22] or a single session [21,23] enhanced working memory. Nikolin et al. suggest that tDCS does not produce substantial improvements in working memory performance in healthy participants [24]. In addition, the information provided by meta-analyses is contradictory and found only partial support for the hypothesis of an enhancement effect of anodal tDCS on working memory performance, noting that reaction times were significantly improved on offline WM tasks [25]. A systematic review found that healthy participants responded significantly faster [8]. A meta-analysis suggests that there is no significant effect of the single session, however, when domains of executive function were analyzed individually, a significant effect of tDCS on working memory performance emerged [26]; other meta-analyses point out that the studies reviewed did not show an effect of tDCS on working memory accuracy [27].

The aim of the study was to test whether the application of anodal active tDCS in healthy older adults produced improvements in verbal memory, learning potential, and executive function. To this end, an active stimulation group was compared with a sham group, both receiving two sessions of stimulation. To analyze the effect on these variables and based on previous research, we applied the stimulation to the left DLPFC. Therefore, the hypothesis is that the application of anodal tDCS will have beneficial effects on cognition in healthy older adults, producing a significant increase in the scores of the active group compared to the sham group.

## 2. Material and Methods

### 2.1. Participants

Recruitment was carried out in a university program for seniors (NAU Gran) at the University of Valencia. To be eligible for inclusion, participants had to be over 65 years of age, have no cognitive impairment, and be able to attend two stimulation sessions on consecutive days. 

A randomized, single-blind, sham-controlled design was used. Once patients were available, allocation to the active anodal stimulation or sham groups was carried out by stratified block randomization. Participants were randomly assigned using a random number system. They were allocated to the groups (active vs. sham) with a 1:1 ratio, with gender as the stratum. Although the researchers were aware of the assigned arm, the patients remained blind to the assignment. 

Initially, 33 healthy older adults were recruited to participate, but two were dropped for not attending all the stimulation sessions. Finally, the sample was composed of 31 healthy older adults (16 women, 15 men) between 65 and 80 years old (*M* = 69.9, *SD* = 4.1) who participated voluntarily and signed the informed consent before starting the study. The Ethical Committee on Human Research of the University of Valencia approved this study.

### 2.2. Instruments

Mini-Mental State Examination [28] was used as an inclusion criterion to rule out participants with cognitive impairment. None of the participants showed cognitive impairment, and the mean for the entire group was 29.8 (*SD* = 0.54, range 28–30). Furthermore, no significant differences were observed at baseline between the stimulation and sham groups (*t*(29) = 0.592, *p* = 0.559).

Complutense Verbal Learning Test (TAVEC) consists of a list of 16 words that are presented to the subject five times in order to evaluate different memory and learning processes [29]. Each trial is scored from 0 to 16, with a maximum total score of 80. The score on the first trial provides a measure of immediate memory or short-term free memory, and the fifth trial assesses the learning of the word list after training it five consecutive times. In addition, the learning potential score is obtained by calculating the difference in the words remembered between the first and fifth trials [30]. An increase in the score indicates a higher level of cognitive function.

Digits forward and digits backward [31] from the Wechsler Intelligence Scale for Adults-III (WAIS-III) were used. The digits forward task assesses immediate recall. It requires the participant to repeat a sequence of numbers in the same order in which they were read. On the digits backward task, the subject must repeat a sequence of numbers in the reverse order of their presentation. The task is used to assess working memory. Both subtests have eight elements with two items each. The test ends when the subject fails on two items in the same element. For each correct item, one point is given, with a maximum score of 16 on each subtest.

Because the assessment of participants was carried out on two consecutive days, it was necessary to select different versions of Complutense Verbal Learning Test (TAVEC) and the WAIS digits subtests to eliminate any bias due to learning.

### 2.3. Procedure

All participants were contacted through the teaching program carried out at the University of Valencia for older adults. The objective of the study was explained to them, and their voluntary participation was requested, after informing them that they would have to attend two consecutive sessions, both appointments being at the same time each day. Once the list of participants had been obtained, an adapted appointment calendar was established, and they were randomly assigned to the active or sham groups.

HDC stimulator (Newronika TM, Milan, Italy) was used to perform non-invasive tDCS with a constant current intensity of 2 mA [9]. Two electrodes with sponges soaked in saline solution (5 × 5 cm) were used. For the positioning of the electrodes, the international 10–20 EEG system was used. The anode was placed on position F7, coinciding with the left dorsolateral prefrontal cortex region, and the cathode was placed on Fp2, coinciding with the right supraorbital area (rSO). The prefrontal cortex is the main brain structure related to executive functions and is made up of the dorsolateral prefrontal cortex (DLPFC), medial, and orbitofrontal/VMPFC regions [32]. Although it is highly interconnected, it has been suggested that the DLPFC, specifically, is more specialized in working memory, a type of executive function [33]. The stimulation time was 20 min, with an initial and final ramp of 30 s so that the participant could adapt to the sensation of the current. The sham group received direct current only on the ramps to generate a sensation of the effect.

The first session began by reviewing the objectives of the study and completing the informed consent. Next, the evaluation protocol was administered. After that, the first session of tDCS stimulation began. The second session began with the application of tDCS stimulation, and after approximately three minutes of stimulation, the evaluation protocol was administered.

### 2.4. Analysis

For the comparison of the groups, *t* tests for independent samples and chi-squared test were used. For the comparison of the baseline and post-treatment measurements of the two groups, mixed ANOVAS with 2 sessions (before versus during intervention; within subjects) × 2 groups (treatment versus control; between subjects) were performed. The data were analyzed with SPSS 21.

Via G*Power to compute a priori statistical power analysis indicated a minimum total sample size of 30 for a power of 0.95 (α = 0.05; 1 −β = 0.95; two groups; 4 measurements, and correlation among repeated measures of 0.5) to detect a Cohen effect size (*f*(v) = 0.84), in an F test of repeated measures for within-between interaction.

## 3. Results

### 3.1. Characteristics of Participants

Participants’ characteristics are listed in Table 1. Age, gender, years of education, and MMSE were not different between the groups (Table 1).

### 3.2. Memory

A mixed ANOVA with 2 sessions (before vs. during intervention; within subjects) × 2 Trials (first vs fifth) × 2 groups (active vs sham; between subjects) was performed on the scores of the TAVEC showed that the main effect of sessions was significant (*F*(1, 29) = 42.76, *p* < 0.001, η^2^*p* = 0.596) as well as the main effect of trials (*F*(1, 29) = 576.42, *p* < 0.001, η^2^*p* = 0.952). However the main effect of the group was not significant (*F*(1, 29) = 1.67, *p* = 0.207, η^2^*p* = 0.054). Regarding interactions the ANOVA showed that sessions by groups interaction was significant (*F*(1, 29) = 13.99, *p* = 0.001, η^2^*p* = 0.325) as well as the interaction of sessions by trials (*F*(1, 29) = 10.36, *p* = 0.003, η^2^*p* = 0.263), but the interaction of sessions by groups was not significant (*F*(1, 29) = 0.24, *p* = 0.624, η^2^*p* = 0.008). Finally, the interaction of sessions by trials by groups was significant (*F*(1, 29) = 13.42, *p* = 0.001, η^2^*p* = 0.316).

To analyze the latter significant interaction, two within subjects’ ANOVAs (2 sessions × 2 trials) were performed, one for the sham group and one for the active group (simple effects tests). Regarding to the sham group (Figure 1) the results showed that main effect of sessions was significant (*F*(1, 14) = 5.09, *p* = 0.041, η^2^*p* = 0.267) as well as the main effect of trials (*F*(1, 14) = 211.88, *p* < 0.001, η^2^*p* = 0.938), but the interaction was not significant (*F*(1, 14) = 0.14, *p* = 0.719, η^2^*p* = 0.010). Regarding to the active group (Figure 1) the results showed that main effect of sessions was significant (*F*(1, 15) = 44.12, *p* < 0.001, η^2^*p* = 0.746) as well as the main effect of trials (*F*(1, 15) = 411.63, *p* < 0.001, η^2^*p* = 0.965), and also the interaction was significant (*F*(1, 15) = 19.29, *p* = 0.001, η^2^*p* = 0.563). A comparison of the pre-test and post-test scores showed that in trial 1 there was no significant change (*p* = 0.110) in the means (M_PRETEST_ = 5.37 vs M_POSTETS_ = 5.81), but in trial 5 there was significant change (*p* < 0.001) in the means (M_PRETEST_ = 11.5 vs. M_POSTETS_ = 14). 

Regarding the direct digits, mixed ANOVAs were performed, obtaining a significant main effect of the session (*F*(1, 29) = 6.32, *p* = 0.018, η^2^*p* = 0.179), but not the group (*F*(1, 29) = 0.12, *p* = 0.126, η^2^*p* = 0.004) or the session × group interaction (*F*(1, 29) = 2.12, *p* = 0.156, η^2^*p* = 0.068; Treatment: *M*_PRETEST_ = 10.06, *M*_POSTETS_ = 11.06; Control: *M*_PRETEST_ = 10.13, *M*_POSTETS_ = 10.4).

### 3.3. Learning Potential

Learning potential was calculated as the difference between trials 5 and 1 on the TAVEC, and mixed ANOVAs were performed for its analysis, obtaining a significant main effect of the session (*F*(1, 29) = 4.43, *p* = 0.044, η^2^*p* = 0.133) and the session × group interaction (*F*(1, 29) = 15.96, *p* < 0.001, η^2^*p* = 0.355), but the group effect was not significant (*F*(1, 29) = 0.03, *p* = 0.847, η^2^*p* = 0,001). The simple effects tests carried out to analyze the significant interaction showed that there were no significant differences between the groups on the pre-test (*F*(1, 29) = 2.51, *p* = 0.124, η^2^*p* = 0.079; *M*_ACTIVE_ = 6.25 vs. *M*_SHAM_ = 7.4). However, significant differences were observed on the post-test (*F*(1, 29) = 4.55, *p* = 0.041, η^2^*p* = 0.136; *M*_ACTIVE_ = 8.18 vs *M*_CONTROL_ = 6.8), due to the increase in the mean of the treatment group. Comparison of the pre-test and post-test scores in each group revealed that, in the control group, there was no significant change in their scores (*F*(1, 29) = 1.73, *p* = 0.199, η^2^*p* = 0.056; *M*_PRETEST_ = 7.4 vs. *M*_POSTETS_ = 6.8), whereas the treatment group showed a significant increase from pre-test to post-test (*F*(1, 29) = 19.24, *p* < 0.001, η^2^*p* = 0.399; *M*_PRETEST_ = 6.25 vs. *M*_POSTETS_ = 8.2) after stimulation application.

### 3.4. Working Memory

Finally, the mixed ANOVA of the inverse digits showed a significant main effect of the session (*F*(1, 29) = 9.57, *p* = 0.004, η^2^*p* = 0.248) and the session × group interaction (*F*(1, 29) = 5.33, *p* = 0,028, η^2^*p* = 0.155), but the main group effect was not significant (*F*(1, 27) < 0.01, *p* = 0.964, η^2^*p* = 0.001). The simple effects tests performed to analyze the significant interaction showed that there were no significant differences between the groups on the pre-test (*F*(1, 29) = 0.43, *p* = 0.513, η^2^*p* = 0.015; *M*_ACTIVE_ = 7.31 vs. *M*_SHAM_ = 7.86) or on the post-test (*F*(1, 29) = 0.75, *p* = 0.395, η^2^*p* = 0.025; *M*_ACTIVE_ = 8.68 vs *M*_SHAM_ = 8.06). The study of the change in each group revealed that, whereas in the control group there was no significant change in the scores between the sessions (*F*(1, 29) = 0.29, *p* = 0.589, η^2^*p* = 0.010) (*M*_PRETEST_ = 7.86 vs. *M*_POSTETS_ = 8.06), the treatment group showed a significant change (*F*(1, 29) = 15.08, *p* = 0.001, η^2^*p* = 0.342) after the application of the stimulation, with an increase in their scores (*M*_PRETEST_ = 7.31 vs. *M*_POSTETS_ = 8.68).

## 4. Discussion

The present study aimed to demonstrate the effectiveness of the application of tDCS in healthy older adults. When comparing the results of the active group and the sham group, differences in working memory and learning potential were found, but no differences in immediate memory were observed.

For immediate memory, in line with previous studies, no significant differences were observed between the treatment and sham groups. It has to be noted that previous research on anodal tDCS on immediate memory found conflicting results. Some studies have demonstrated the effects of tDCS on short-term memory, however, these have been in a clinical population; for example, short-term facilitation effects were observed in Alzheimer’s disease [34]. Additionally, compared to baseline and sham groups, MCI patients showed better performance on neuropsychological assessments exploring immediate verbal memory after anodal stimulation [35]. In addition, the combination of a tDCS treatment coupled with a working memory task compared to cognitive training alone showed a greater increase in immediate recall and verbal fluency after active treatment compared to sham [36]. 

However, after applying anodal tDCS to the left DLPFC to explore the modulatory effect online with healthy subjects some authors found no significant effects on short-term memory tasks suggesting that the timing of tDCS administration could have an influence on the effects [37]. This supports the results of previous studies which systematically compared online and offline stimulation in other domains and found prominent offline effects [38] similar to those obtained in this study. One possible explanation of this difference is that the temporal specificity of tDCS that varies as a function of the involvement of the stimulated brain region during a specific stage of processing and associated cognitive functions [39]. Another study suggests that tDCS does not improve immediate recall, because it only alters plasticity during learning [40], and perhaps during consolidation, leading to stronger and more persistent memories [13]. Specifically, when learning new material, the excitatory-inhibitory balance in neuronal networks is disturbed, and the application of anodal tDCS in this phase seems to lead to beneficial effects and enhance the learning process, as some studies have demonstrated [8,25]. Finally, a possible justification for this lack of evidence is that there is no particular structure specialized in the function of memory because it is actually stored in patterns of connectivity in different areas [39]. In addition, various regions, including the DLPFC, are coordinated in the processing, encoding, and retrieval of verbal information [41,42,43]. In this way, targeting a single structure can be a limitation of the study [44]. 

Regarding the learning potential, the hypothesis that the group that received tDCS would have a higher rate of correctly remembered items compared to the sham group was confirmed. Perceval et al. studied verbal associative learning and its long-term effects in healthy older adults [45]. Improvements in learning and memory capacity were found, but only in the group of older adults whose initial learning performance was lower. Thus, these results show that the short-term and long-term effects of tDCS depend on the baseline cognitive state and particularly benefit older adults who may need help. 

Neuroplasticity is based on cortical and neuronal excitability and stimulating excitability with tDCS enhances cognitive processes [46], through the fortification of neural connections [47,48] and, therefore, a reconfiguration of brain networks [49]. For example, Huo et al. [18] found that tDCS was effective when stimulating the DLPFC because neural changes in the prefrontal cortex resulted in better memory performance.

Regarding working memory, our results are consistent with other studies showing improvements in WM performance with the tDCS technique [50,51,52,53], which means that tDCS can help to improve this cognitive ability. Effects of tDCS combined with working memory training have been shown to extend and increase training gains [2,22,51]. Thus, tDCS seems to be a useful intervention technique to reverse the typical working memory decline observed in aging [52]. 

However, more recent meta-analyses have concluded that the effects are small [25] or partial [54]. Thus, more empirical studies are needed on the impact of tDCS on functional connectivity of the working memory network, in order to harness and optimize tDCS as a treatment approach for cognitive decline in older adults [52]. 

Although there is still no consensus about the reason for the efficacy of tDCS for working memory, the most widely accepted explanation today is that anodal stimulation of the left DLPFC may increase the effectiveness of WM training and be useful when applied before or during WM tests [55]. Moreover, tDCS can improve WM because it modulates excitability and cerebral cortical activity by transmitting a weak electrical current in the brain [22,54]. A more recent study explained that tDCS selectively modulates frontal functional connectivity of the working memory in older adults [52]. Different lines of studies, including functional imaging, indicate that working memory processes are mediated through distributed subcortical and cortical networks that include different subregions of the prefrontal cortex (PFC) [56]. The prefrontal cortex is selectively involved in linking sensory information with task-relevant information about goals, actions, rules, and strategies to achieve these goals. These findings suggest that the neural networks supporting working memory include several crucial hubs such as the dorsolateral prefrontal cortex, the anterior cingulate cortex, and the posterior parietal cortex [57]. Testing the different mechanisms that influence the effectiveness of tDCS should be a future goal.

### Limitations and Future Lines

Some limitations of this study must be acknowledged. First, the intervention consists of only two sessions, which may not be sufficient. Second, cognitive functions were evaluated before and during the intervention to determine whether it had significant effects, but there was no follow-up to find out whether the effects are sustained over time. Third, it should be noted that by placing the cathode at Fp2 the current could cause unwanted inhibition of some nearby areas, in contrast, some authors point out that extracranial cathodes increase the facilitation of some functions compared to cranial cathodes as supraorbital zone [2]. Finally, it should be noted that the participants were included in a university training program and were therefore healthy older adults with an interest in active learning and therefore motivated to continue using their cognitive functions.

For future lines of research, we propose applying this same intervention to different populations. Because its efficacy has been shown in a healthy population for various cognitive functions, it is plausible to imagine that it could also work in cases of incipient cognitive deterioration. Thus, it could serve as an early intervention or when the disease has already manifested itself and achieve an improvement in the quality of life. An intervention protocol that includes a larger number of sessions could also be designed. Furthermore, to find out whether the effects are sustained in the long term, future studies should carry out a follow-up evaluation sometime after the intervention. Finally, another possible line of future research concerns the intensity of stimulation. Some research has suggested that an enhancement of tDCS intensity does not necessarily increase the efficacy of stimulation but might also shift the direction of excitability alterations [6]. It might be interesting to use the same design to apply an intensity of 1 mA in order to compare the results of the two active groups and see if the gain is really a function of intensity.

## 5. Conclusions

In sum, the efficacy of tDCS was observed for working memory and potential learning variables, whereas for immediate memory no significant differences were found after the intervention. Thus, the results presented show that tDCS is a non-invasive and safe tool to enhance cognitive processes in healthy older adults interested in maintaining cognitive function. However, because the results in the literature are diverse, more evidence is needed about its effectiveness in relation to the stimulated brain areas, in order to reach a consensus about why stimulating these specific areas improves some cognitive processes and not others.

## Figures and Tables

**Figure 1 ijerph-19-12716-f001:**
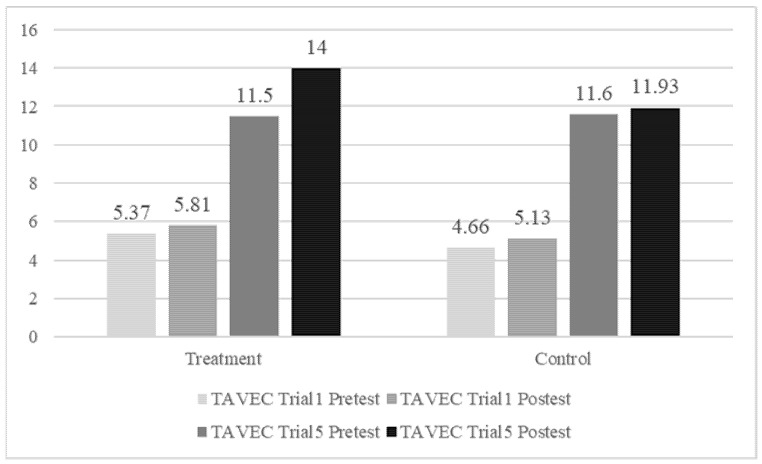
Results of trials 1 and 5 in the TAVEC test.

**Table 1 ijerph-19-12716-t001:** Characteristics of participants at baseline (mean ± SD).

	Active Group (*n* = 16)	Sham Group (*n* = 15)	*p*-Value
Age	69.8 ± 3.4	70.13 ± 4.7	0.217 ^a^
Gender (female/male)	8/8	8/7	0.987 ^b^
Years of Education	12.25 ± 3.29	11.67 ± 3.3	0.627 ^a^
MMSE	29.75 ± 0.57	29.86 ± 0.51	0.559 ^a^

Statistical tests: (a) Student’s *t*-test for independent samples, (b) Chi-squared test.

## Data Availability

The data presented in this study are available on request to the authors. The data are not publicly available due to privacy reasons.

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
