# Peer review of "Enhancing Immediate Memory, Potential Learning, and Working Memory with Transcranial Direct Current Stimulation in Healthy Older Adults"

_ijerph, 2022, doi:10.3390/ijerph191912716_

Round 1
Reviewer 1 Report
Thank you for the opportunity to review your manuscript entitled " Enhancing immediate memory, potential learning, and working memory with transcranial direct current stimulation in healthy older adults". The method looks suitable, but some presentation content confused me. Please see detail comments below.
1. Did this study register study protocol in advance on any website? Please clarify.
2. IRB number should be provided.
3. It is very important to explain why author design two-session protocol (not longer sessions or just one session)? Please provide adequate reason and your hypothesis. In addition, the inter-session period is 24 hours or just “next day” (variable time).
4. Line 148 to 150: “Because the assessment of participants was carried out on two consecutive days, it was necessary to select different versions of these two tests to eliminate any bias due to learning”. Why are “these two tests”? Does it mean TAVEC and WAIS? Please clarify it.
5. Did author investigate the effectiveness of blinding? Please clarify
6. How you chose F7 and Fp2? Based on 10-20 EEG, Beam, 5-cm rule? Please clarity.
7. Did you restrict your participant as right-handed? If not, please explain it?
8. Author should provide Table for detail demographic characteristics between groups to make more clinically relevant.
9. To be honest, I am confused about the result part, because so many information was provided. I suggest author add subtitled before each paragraph to make reader friendly.
10. I am confused about Figure 1. Did all the results of Trials 1 (pre- and post-) and Trials 2 (pre- and post-) are based on Session 1 or 2, or combined both 1 and 2? Please clarify it.
11. Author mentioned WAIS-III in the method section, but no information regarding WAIS in the results provided? Is it correct?
12. It is important to describe the mean age, gender, education in the results.
Reviewer 2 Report
The manuscript present a research article on the use of tDCS in normal elderly people to demonstrate the efficacy of such method on some aspects of memory Overall the article is well presented, the style and the order of section and subsection are clear, the English is well curated, and the effort of the authors is highly valuable. However some major issues emerged after reading the manuscript
MAJOR ISSUES
There are two most limiting factors to the present study. The first is the use of the MMSE as a screening tool (see later).
The second is the lack of a list of variables that could have an affect on the use and the performance of the tDCS. Specifically, it is well known that tDCS is influenced by a relevant number of possible confounders, like: handedness, pharmacological effects, genetics, regular practice of cognitive exercise, endogenous brain oscillations (EEG coupling), shape of the head, and electrode brain distance, and daytime of application (see for example: Determinants of the induction of cortical plasticity by non-invasive brain stimulation in healthy subjects. J. Physiol., 588 (2010)). Without taking into account these (and others) possible confounders, it is difficult to reproduce the results, and consequently to draw a conclusion.
Line 37: the sentence “prevention method or minimizer of the normal cognitive deterioration” is not scientifically sound. There are no evidence to support a strong statement like that.
Introduction section: although very well reviewed, to be more complete, the authors should define terms like “anodal tDCS”, “cathodal tDCS” (with differences in procedure and results), ”online tasks” and “offline tasks”.
Line 107: recruited participants are people over 65 years of age, who “have no cognitive impairment”. It is not clear how the participants were recruited. In line 152 it is stated that participants were contacted through the teaching program carried out at the University, but the screening for excluding a diagnosis of Alzheimer’s disease or Mild Cognitive Impairment is based only on the MMSE (line 127) which is actually the most used but with out-of-date. The latest research and the authors should be encouraged to use other screening tool which are superior to MMSE (like MoCA, see: Pinto, T. C., Machado, L., Bulgacov, T. M., Rodrigues-Júnior, A. L., Costa, M. L., Ximenes, R. C., & Sougey, E. B. (2019). Is the Montreal Cognitive Assessment (MoCA) screening superior to the Mini-Mental State Examination (MMSE) in the detection of mild cognitive impairment (MCI) and Alzheimer’s Disease (AD) in the elderly?. International Psychogeriatrics, 31(4), 491-504), or, better, to use different neuropsychological tests to assess different aspects of cognition, before excluding a diagnosis of AD or MCI.
Line 125: the academic level of the included patients was not stated. However given the mean score to MMSE (line 129) it is presumably a high level of scholarly. This should be considered as a possible limitation of the study, or vice versa, an indicator that the study was conducted in elderly people, with high level of education, and intact memory.
Line 138-139: the sentence “an increase in the score indicates a higher level of cognitive plasticity” needs a clarification, since the term “plasticity” has its special meaning in terms of cognition.
Line 132: why the authors decided to use the TAVEC test and the forward/backward digit span test? Why not using a more common test, or not using a more extensive test battery?
Line 159: high intensity (2.0 mA) and long duration stimulation (20 min), can partially reverse the effects of tDCS (see for example: Partially non-linear stimulation intensity-dependent effects of direct current stimulation on motor cortex excitability in humans. J. Physiol., 591 (2013)). The choice of such parameters should be therefore justify by the authors.
Line 207-209: the sentence should be removed from the “result” section, since it is a “conclusion”.
MINOR ISSUES
Throughout the manuscript, sometimes the control group is named “placebo-“, and sometimes “sham-“. The second is the correct form in cases of tDCS.
Line 69-70: the sentence is not clear, and should be re-written.
Line 76-80: it is not clear if the authors are saying that tDCS has never been used in people affected by Alzheimer’s disease and Mild Cognitive Impairment. If this is the case, the authors should note that there are an increasing number of studies of tDCS application in AD and MCI (e.g.
Transcranial direct current stimulation improves recognition memory in Alzheimer disease. Neurology, 71 (2008).
Temporal cortex direct current stimulation enhances performance on a visual recognition memory task in Alzheimer disease. J. Neurol. Neurosurg. Psychiatry, 80 (2009).
Anodal tDCS during face-name associations memory training in Alzheimer’s patients. Front. Aging Neurosci. (2014)).
Line 140: a phrasal verb is missing.
Round 2
Reviewer 2 Report
The authors answered every suggestion of the previous review, and the manuscript has improved in quality.